# *CDKN2A* Homozygous Deletion Is a Stronger Predictor of Outcome than *IDH1/2*-Mutation in CNS WHO Grade 4 Gliomas

**DOI:** 10.3390/biomedicines12102256

**Published:** 2024-10-04

**Authors:** Sang Hyuk Lee, Tae Gyu Kim, Kyeong Hwa Ryu, Seok Hyun Kim, Young Zoon Kim

**Affiliations:** 1Division of Neuro Oncology and Department of Neurosurgery, Samsung Changwon Hospital, Sungkyunkwan University School of Medicine, Changwon 51353, Republic of Korea; shlee858@naver.com; 2Department of Radiation Oncology, Samsung Changwon Hospital, Sungkyunkwan University School of Medicine, Changwon 51353, Republic of Korea; tg1.kim@samsung.com; 3Department of Radiology, Samsung Changwon Hospital, Sungkyunkwan University School of Medicine, Changwon 51353, Republic of Korea; ryukh0329@gmail.com; 4Division of Hematology and Medical Oncology, Department of Internal Medicine, Samsung Changwon Hospital, Sungkyunkwan University School of Medicine, Changwon 51353, Republic of Korea; tjrgus1@hanmail.net

**Keywords:** glioma, *CDKN2A* deletion, *IDH* mutation, glioblastoma, astrocytoma, prognosis

## Abstract

**Background**: We primarily investigated the prognostic role of *CDKN2A* homozygous deletion in CNS WHO grade 4 gliomas. Additionally, we plan to examine traditional prognostic factors for grade 4 gliomas and validate the findings. **Materials**: We conducted a retrospective analysis of the glioma cohorts at our institute. We reviewed medical records spanning a 15-year period and examined pathological slides for an updated diagnosis according to the 2021 WHO classification of CNS tumors. We examined the *IDH1/2* mutation and *CDKN2A* deletion using NGS analysis with ONCOaccuPanel^®^. Further, we examined traditional prognostic factors, including age, WHO performance status, extent of resection, and *MGMT* promoter methylation status. **Results**: The mean follow-up duration was 27.5 months (range: 4.1–43.5 months) and mean overall survival (OS) was 20.7 months (SD, ±1.759). After the exclusion of six patients with a poor status of pathologic samples, a total of 136 glioblastoma cases diagnosed by previous WHO classification criteria were newly classified into 29 (21.3%) astrocytoma, *IDH*-mutant, and CNS WHO grade 4 cases, and 107 (78.7%) glioblastoma, *IDH*-wildtype, and CNS WHO grade 4 cases. Among them, 61 (56.0%) had *CDKN2A* deletions. The high-risk group with *CDKN2A* deletion regardless of *IDH1/2* mutation had a mean OS of 16.65 months (SD, ±1.554), the intermediate-risk group without *CDKN2A* deletion and with *IDH1/2* mutation had a mean OS of 21.85 months (SD, ±2.082), and the low-risk group without CDKN2A deletion and with *IDH1/2* mutation had a mean OS of 33.38 months (SD, ±2.946). Multifactor analysis showed that age (≥50 years vs. <50 years; HR 4.645), WHO performance (0, 1 vs. 2; HR 5.002), extent of resection (gross total resection vs. others; HR 5.528), *MGMT* promoter methylation, (methylated vs. unmethylated; HR 5.078), *IDH1/2* mutation (mutant vs. wildtype; HR 6.352), and *CDKN2A* deletion (absence vs. presence; HR 13.454) were associated with OS independently. **Conclusions**: The present study suggests that *CDKN2A* deletion plays a powerful prognostic role in CNS WHO grade 4 gliomas. Even if CNS WHO grade 4 gliomas have mutant *IDH1/2*, they may have poor clinical outcomes because of *CDKN2A* deletion.

## 1. Introduction

In terms of the nomenclature and grading of common adult-type diffuse astrocytic gliomas, the 2016 World Health Organization (WHO) Classification of Tumors of the Central Nervous System (CNS) assigned isocitrate dehydrogenase (*IDH)*-mutant diffuse astrocytic tumors to three different tumor types (diffuse astrocytoma, anaplastic astrocytoma, and glioblastoma) depending on histologic parameters and immunohistochemical features [1]. In the current revised fourth edition of these classifications, all *IDH*-mutant diffuse astrocytic tumors are considered a single type (astrocytoma, *IDH*-mutant) and are graded as CNS WHO grade 2, 3, or 4 [2]. Moreover, grading is no longer entirely histological because the finding of a cyclin-dependent kinase inhibitor *(CDKN)2A* and/or *CDKN2B* homozygous deletion results in a CNS WHO grade of 4, even in the absence of microvascular proliferation or necrosis [2]. *CDKN2A/B* homozygous deletions were previously identified as a negative prognostic factor in *IDH*-mutant astrocytomas in update 5 of the Consortium to Inform Molecular and Practical Approaches to CNS Tumor Taxonomy. *IDH*-mutant astrocytoma grade 4 was defined as a diffusely infiltrative astrocytic glioma with an *IDH1* or *IDH2* mutation that exhibited microvascular proliferation or necrosis, *CDKN2A/B* homozygous deletion, or any combination of these features [3]. Consequently, many diagnostic algorithms for diffuse adult gliomas include *CDKN2A/B* deletions [4].

*CDKN2A* is a tumor suppressor gene located on chromosome 9p21 that encodes cell cycle inhibitor protein p16 [5]. Mutations or deletions in the *CDKN2* family of tumor suppressor genes are present in 30–80% of gliomas [6]. *CDKN2A/B* are tumor suppressor genes that encode p16^CDKN2A^ and p15^INK4B^ proteins, respectively, which can inhibit the activity of CDK kinases and regulate the G1 cell cycle; inactivation of *CDKN2A/B* may lead to uncontrolled cell growth and proliferation [7]. Previous studies have reported that the homozygous deletion of *CDKN2A* is associated with high-grade disease, particularly glioblastoma, and *CDKN2A/B* risk variants appear to have a general effect on tumor risk [8]. However, recent studies have shown that *CDKN2A/B* loss points towards a more aggressive phenotype, even when derived from low-grade gliomas [9]. In patients with *IDH*-mutated astrocytomas, the presence of *CDKN2A/B* homozygous deletion leads to clinical behavior consistent with that of CNS WHO grade 4 gliomas [3]. Moreover, patients with *CDKN2A/B* homozygous deletion tumors have a worse prognosis than those without deletions [10]. Progression-free survival (PFS) and overall survival (OS) are significantly shorter in patients with *IDH*-mutated low-grade gliomas with homozygous deletion of *CDKN2A* [11,12]. These results led to the integration of *CDKN2A/B* status into the 2021 WHO CNS classification [2].

As described above, many roles for *CDKN2A/B* homozygous deletion as a negative prognostic factor have been reported for *IDH*-mutant low-grade gliomas. However, until now, only a few comprehensive studies have reported on the effect of *CDKN2A/B* homozygous deletion on the prognosis of CNS WHO grade 4 gliomas of the *IDH* wildtype. Here, we investigated the prognostic role of *CDKN2A* homozygous deletion in WHO-grade 4 CNS gliomas. Additionally, we plan to examine traditional prognostic factors for CNS WHO grade 4 gliomas and validate the results.

## 2. Materials and Methods

### 2.1. Patients and Sample Collection

This translational cohort study was conducted between January 2006 and December 2023 using formalin-fixed, paraffin-embedded (FFPE) tissue specimens obtained from patients with WHO grade 4 CNS gliomas via biopsy or surgical resection at our institute. In total, 151 patients were histopathologically diagnosed with CNS WHO Health Organization grade 4 gliomas following surgical resection or biopsy. Of these, 142 patients who were treated at our institute during the entire disease process and followed up until death were included in this study. Our institute is the only tertiary medical center that covers the surrounding population of 1.5 million and is equipped with advanced medical equipment and facilities; therefore, there has been little outflow of patients to other areas or hospitals. Patients with histories of other cancers were excluded from the study.

Histological samples were obtained from the Department of Pathology archives of our institute. All hematoxylin and eosin-stained slides were reviewed by two pathologists (Dr. Lee EH and Dr. Kim KS, Samsung Changwon Hospital, Changwon, Republic of Korea) using the 2021 revision of the WHO classification of CNS tumors [2]. The pathologists were blinded to the clinical and pathological parameters. Samples in poor condition were excluded if the tumor was almost entirely necrotized or if its contribution to the section was less than 80%. Patients with insufficient medical data were excluded from the analysis.

### 2.2. Clinical Data

The epidemiological characteristics (including sex, age at initial diagnosis, and World Health Organization performance status), extent of resection, recursive partitioning analysis (RPA) classification, type of postoperative adjuvant treatment, duration of follow-up, and dates of recurrence and death were retrospectively reviewed from the medical records. Additionally, salvage treatment modalities after progression were examined. The WHO performance status was classified according to the content defined in the existing literature [13]. RPA class was determined using the modified Radiation Therapy Oncology Group (RTOG) method, and the RPA score was assessed based on age, performance status, extent of resection, and neurologic function [14]. For example, patients with RPA class III are defined as those under the age of 50 with a KPS score of ≥90, and patients with RPA class IV are those under the age of 50 with a KPS score of <90, or those over the age of 50 with a KPS score of ≥70 and have undergone gross total resection (GTR), and patients with RPA class V are defined as those over the age 50 with a KPS score of ≥70 but have undergone subtotal resection (STR) or biopsy only, or those over the age of 50 and with a KPS score of <70 [14].

The radiological characteristics of the brain lesions were evaluated using conventional magnetic resonance imaging (MRI) with gadolinium (Gd) enhancement, MR perfusion, and MR spectroscopy at the time of initial diagnosis. Elements for radiological features, such as peritumoral edema and extents of resistance, were evaluated by the established protocols of this institution that have been published in the literature [15]. Measurements to evaluate response to various treatments were evaluated according to Radiologic Assessment in Neuro Oncology (RANO) criteria [16]. Radiological evaluations were performed by two neuroradiologists (Dr. Ryu KH and Dr. Byun HS, Samsung Changwon Hospital, Changwon, Republic of Korea) who were blinded to the clinical and pathological parameters.

Routine analysis of diagnostic markers was performed at the time of the initial histopathological diagnosis. Cellularity, cellular pleomorphism, mitotic count, microvascular proliferation, cellular necrosis, and *MGMT* gene promoter methylation were evaluated from pathological reports. MGMT gene promoter methylation has been tested using the pyrosequencing method as a quantitative tool in our institute [15] instead of the methylation-specific polymerase chain reaction (PCR). The main analyses converted individual C/T ratio data into mean values of the five CpGs analyzed in a gene segment. The percentage of the mean value of five CpGs were considered methylated if the percentage was >9%, which is a widely used reference in the literature [15]. Additionally, for the molecular diagnosis of CNS WHO grade 4 gliomas according to the new 2021 WHO classification of CNS tumors [2], the presence of *1p19q* co-deletions and *IDH1* or *2* mutations was investigated to differentiate between WHO CNS grade 4 astrocytoma, *IDH*-mutant, and glioblastoma, *IDH*-wildtype, using the FFPE samples of the tumors, including those obtained before the 2021 new version of classification.

### 2.3. Next-Generation Sequencing for Genetic Alteration

For the detection of *CDKN2A/B, IDH1* or *2* mutation, and *TERT* promoter mutation, ONCOaccuPanel^®^ (NGeneBio, Seoul, Republic of Korea) on the Illumina MiSeq platform was used for NGS. ONCOaccuPanel is a hybridization capture-based DNA panel that detects somatic mutations and copy number alterations in 323 key cancer genes, and fusions of 17 genes in solid tumors. ONCOaccuPanel DNA probes were designed for the targeted sequencing of all exons and selected introns of 225 genes and partial exons of 98 genes (a total of 323 genes) (Appendix A). The preparation process including DNA extraction, DNA purification, and DNA quantification were followed by institutional protocol [17]. Other processes for NGS analysis such as library preparation, determination of coverage requirements, and target region coverage were performed as previously described [17]. The *IDH* mutation was defined in this analysis as single-amino-acid missense mutation in *IDH1* at arginine 132 (R132) or the analogous residue in *IDH2* (R140 or R172) by the single-nucleotide polymorphism (SNP) and insertion/deletion polymorphism (INDEL) analysis of NGS.

### 2.4. Survival Analysis and Statistical Analysis

Medical records of the patients’ clinical histories and radiographic reports were analyzed. The date of death was recorded. OS was defined as the time from the date of diagnosis of CNS WHO grade 4 gliomas until death. PFS was defined as the time from the date of diagnosis until the detection of progression on follow-up MRI. Progression was defined as the presence of a new enhancing tumor mass at the resected site as judged on the first postoperative MRI. The date of the biopsy or surgical resection of the tumor was recorded as the date of diagnosis. Statistical analyses were performed using SPSS ver. 29.0.2.0 (IBM Corp., Armonk, NY, USA). In addition, statistical comparative analysis between the two groups or survival analysis such as of PFS and OS were performed by conventional statistical techniques used in existing research results [15]. *p*-values < 0.05 were considered statistically significant.

## 3. Results

### 3.1. Clinical and Genetic Characteristics of Patients

Among the 142 patients newly diagnosed with glioblastoma between January 2006 and December 2023, 136 (84 males and 52 females) were included in this study (Table 1). The remaining six patients (4.2%) were excluded for the following reasons: the tissues were almost entirely necrotized in four patients, the tumor contribution to each section was less than 80% in one patient, and medical data were insufficient in two cases.

The mean age of these patients at the time of diagnosis was 55.9 years (range 29.4–81.6 years). Fifty-five patients (40.4%) were fully active and able to perform all pre-disease activities without restriction (WHO performance status 0), whereas 81 patients (59.6%) demonstrated restricted strenuous physical activity in daily life (WHO performance status 1 or 2). Sixty-seven patients (49.3%) underwent GTR, 52 (38.2%) underwent STR, and 17 (12.5%) were diagnosed with glioblastoma after biopsy only. Thirty-two patients (23.5%) were classified as RPA class III; 79 (58.1%) as RPA class IV; and 18 (18.4%) as RPA class V (Table 1).

The *MGMT* promoter was methylated in 86 (63.2%) patients and unmethylated in 50 (36.8%), as analyzed by quantitative pyrosequencing. EGFR amplification was detected in 95 patients (69.9%), but not in 41 patients (30.1%) using NGS analysis. *TERT* promoter mutations were found in 71 patients (52.2%), but not in 65 patients (47.8%), by NGS analysis (Table 1).

Out of 29 patients (21.3%) with *IDH* mutation, 28 patients (20.6%) had *IDH1* mutations, and 1 patient (0.7%) had *IDH2* mutations; they were diagnosed with CNS WHO grade 4 astrocytoma, *IDH*-mutant, instead of glioblastoma. Another 107 patients (78.7%) without *IDH1* or *2* mutations were diagnosed with CNS WHO grade 4 glioblastoma, *IDH*-wildtype. *CDKN2A* homozygous deletion was found in 75 patients (55.1%), but not in 61 patients (47.8%), by NGS analysis (Table 1). *CDKN2B* deletions were not detected in any of the samples.

For postoperative adjuvant treatment, 35 patients (25.7%) underwent nitrosourea-based combination chemotherapy with or without radiotherapy, whereas 101 patients (74.3%) underwent concurrent chemoradiotherapy with temozolomide. After tumor progression, 74 patients (54.4%) underwent a second resection, 63 (46.3%) were treated with repeated irradiation, 83 (61.0%) received salvage chemotherapy or targeted therapy using bevacizumab, and 17 (12.5%) received best supportive care only (Table 1).

Comparative analysis of the clinical characteristics in the group with *IDH1* or *2* mutation and the group without *IDH1* or *2* mutation, although not statistically significant, showed a tendency toward younger age (*p* = 0.153) and better RPA class (*p* = 0.183) in the group with mutations than in the group without mutations (Table 2). Other clinical factors showed no significant differences between the two groups.

Similarly, clinical characteristics were analyzed in the two patient groups according to the presence or absence of *CDKN2A* homozygous deletion; although it was not statistically significant, the patient group with *CDKN2A* homozygous deletion had a higher age (*p* = 0.054) and a higher rate of *TERT* promoter mutation (*p* = 0.103) than the patient group without *CDKN2A* homozygous deletion (Table 3). Other clinical factors showed no significant differences between the two groups.

### 3.2. Univariate Analysis of Factors Predicting Progression-Free Survival

The mean follow-up duration was 27.5 months (ranging from 4.1 to 43.5 months). During follow-up, 129 patients (94.8%) experienced progression, and only 7 patients (5.2%) remained stable. The mean PFS was 10.3 months (standard deviation [SD], ±0.624). In terms of surgical extent, patients who underwent STR (hazard ratio [HR], 9.514; 95% confidence interval [CI], 7.008–12.021) and GTR (HR, 41.251; 95% CI, 34.897–47.605) had significantly longer PFS than those who underwent biopsy only. Patients with RPA class IV (HR, 6.288; 95% CI, 5.543–7.033) and III (HR, 8.648; 95% CI, 6.893–10.403) had significantly longer PFS than those with class V. Patients with a methylated MGMT gene promoter (HR, 7.330; 95% CI, 6.134–8.526) had significantly longer PFS than those with an unmethylated MGMT gene promoter. Patients without EGFR amplification (HR, 14.536; 95% CI, 12.085–16.987) had significantly longer PFS than those with EGFR amplification. Patients without *TERT* promoter mutations (HR, 17.490; 95% CI, 14.922–20.058) had a significantly longer PFS than those with *TERT* promoter mutations. Patients without *CDKN2A* homozygous deletion (HR, 33.218; 95% CI, 30.085–36.351) had a significantly longer PFS than those with *CDKN2A* homozygous deletion (Table 4). These findings were also observed in the Kaplan–Meier survival analysis (Appendix A).

### 3.3. Univariate Analysis of Factors Predicting Overall Survival

The mean OS was 20.7 months (SD, ±1.759). During follow-up, 115 patients (84.6%) succumbed to the disease and only 21 patients (15.4%) were alive. Patients aged < 50 years (HR, 6.326; 95% CI, 4.228–8.424) had statistically longer OS than those aged ≥ 50 years. In terms of performance status, patients with a WHO performance score of 0 (HR, 22.012; 95% CI, 15.321–28.703) and 1 (HR, 10254; 95% CI, 6.925–13.583) had significantly longer OS than those with a WHO performance score of 2. In terms of surgical extent, patients who underwent STR (HR, 4.595; 95% CI, 1.889–7.301) and GTR (HR, 7.980; CI, 4.669–11.291) had significantly longer OS than those who underwent biopsy only. Patients with RPA class IV (HR, 11.620; 95% CI, 7.074–16.166) and III (HR, 23.161; 95% CI, 17.426–28.896) had significantly longer OS than those with class V. Patients with methylated MGMT gene promoters (HR, 6.306; 95% CI, 4.653–7.959) had significantly longer OS than those with unmethylated MGMT gene promoters. Patients without *CDKN2A* homozygous deletion (HR, 11.129; 95% CI, 7.048–15.211) had a significantly longer OS than those with *CDKN2A* homozygous deletion. Patients with *IDH1/2* mutations (HR, 5.794; 95% CI, 3.185–8.403) had a significantly longer OS than those without *IDH1/2* mutations. However, other clinical factors such as sex (*p* = 0.467), EGFR amplification (*p* = 0.320), *TERT* promoter mutation (*p* = 0.208), and therapeutic modalities of postoperative adjuvant treatment (*p* = 0.063) were not associated with OS (Table 4). These findings were also observed in the Kaplan–Meier survival analysis (Appendix A).

### 3.4. Combined Role of CDKN2A Homozygous Deletion and IDH1/2 Mutation

Four subgroups were formed according to *CDKN2A* deletion and *IDH1/2* mutation as follows: Group A, *CDKN2A* deletion and *IDH1/2* wildtype; Group B, *CDKN2A* deletion and *IDH1/2* mutant; Group C, *CDKN2A* intact and *IDH1/2* wildtype; and Group D, *CDKN2A* intact and *IDH1/2* mutant. The mean PFS was 8.53 months (SD, ±0.669) in Group A, 9.72 months (SD, ±0.721) in Group B, 11.60 months (SD, ±0.925) in Group C, and 15.25 months (SD, ±1.492) in Group D (Appendix A). Also, mean OS was 15.63 months (SD, ±1.521) in Group A, 19.67 months (SD, ±1.875) in Group B, 22.63 months (SD, ±2.234) in Group C, and 33.38 months (SD, ±2.946) in Group D (Appendix A). In this subgroup analysis, even patients with *IDH1/2* mutation (Group B), previously known as a better prognostic factor for glioma, had shorter PFS and OS than patients with the *IDH1/2* wildtype if accompanied by *CDKN2A* deletion (Group C). *CDKN2A* deletion was associated with short PFS and OS with or without *IDH1/2* mutation, but *IDH1/2* mutation did not influence PFS or OS in patients with *CDKN2A* deletion (Appendix A). In addition, the hazard ratio of *CDKN2A* deletion was relatively higher than that of *IDH1/2* mutation for PFS (33.218 vs. 3.349, Table 4) and OS (11.794 vs. 5.794, Table 4).

As described above, based on the finding that *CDKN2A* deletion affects the prognosis more strongly than *IDH1/2* mutation in CNS WHO grade 4 glioma, the cohorts were divided into three subgroups as follows: if there is *CDKN2A* deletion regardless of *IDH1/2* mutation, it is classified as the ‘*high risk group*’, if there is no *CDKN2A* deletion with the *IDH1/2* wildtype, it is classified as the ‘*intermediate risk group*’, and if there is no *CDKN2A* deletion with *IDH1/2* mutation, it is classified as the ‘*low risk group’* (Appendix A). The mean PFS was 8.75 months (SD, ±0.694) in the high-risk group, 11.02 months (SD, ±0.956) in the intermediate-risk group, and 15.25 months (SD, ±1.492) in the low-risk group (Table 5). The mean OS was 16.65 months (SD, ±1.554) in the high-risk group, 21.85 months (SD, ±2.082) in the intermediate-risk group, and 33.38 months (SD, ±2.946) in the low-risk group (Table 5). The Kaplan–Meier survival curve analysis showed the same results (Appendix A).

### 3.5. Multivariate Analysis of Factors Predicting Progression-Free Survival

In multivariate analysis using the Cox-regression model, the following factors were independently associated with PFS: (1) extent of resection (HR 11.651, 95% CI 8.755–14.547 in GTR vs. biopsy; HR 9.323, 95% CI 7.285–11.361 in GTR vs. STR; HR 8.609, 95% CI 6.238–10.979 in STR vs. biopsy), (2) RPA class (HR 5.382, 95% CI 3.894–6.869 in III vs. V; HR 4.611, 95% CI 1.996–7.226 in IV vs. V), (3) MGMT gene promoter methylation vs. unmethylation (HR 6.989, 95% CI 5.198–8.779), (4) EGFR amplification vs. non-amplification (HR 9.658, 95% CI 8.113–11.203), (5) *TERT* promoter mutation vs. wildtype (HR 13.077, 95% CI 10.840–15.314), and (6) *CDKN2A* deletion (HR 21.361, 95% CI 18.651–24.071) (Table 6).

### 3.6. Multivariate Analysis of Factors Predicting Overall Survival

In multivariate analysis using the Cox regression model, the following factors were independently associated with OS: (1) age < 50 years vs. ≥50 years (HR 4.645, 95% CI 2.865–6.425), (2) WHO performance status (HR 3.817, 95% CI 2.436–5.198 in 0 vs. 1; HR 5.002, 95% CI 3.756–6.248 in 0 vs. 2; HR 3.663, 95% CI 1.492–5.834 in 1 vs. 2, (3) extent of resection (HR 8.075, 95% CI 5.837–10.313 in GTR vs. biopsy; HR 5.528, 95% CI 3.840–7.216 in GTR vs. STR), (4) RPA class (HR 3.992, 95% CI 2.008–5.976 in III vs. IV; HR 6.773, 95% CI 4.259–9.287 in III vs. V; HR 5.019, 95% CI 3.890–6.148 in IV vs. V), (5) MGMT gene promoter methylation vs. unmethylation (HR 5.078, 95% CI 3.694–6.462), (6) *CDKN2A* deletion vs. *CDKN2A* intact (HR 13.454, 95% CI 10.268–16.639), and (7) *IDH1/2* mutation vs. *IDH1/2* wildtype (HR 6.352, 95% CI 5.079–7.625) (Table 6). However, factors that tended to be associated with OS in the univariate analysis, such as EGFR amplification, *TERT* promoter mutation, and therapeutic modalities of postoperative adjuvant therapy, were not associated with OS (Table 6).

## 4. Discussion

The primary aim of the present study was to investigate the role of *CDKN2A/B* deletion as a poor prognostic factor for CNS WHO grade 4 gliomas. Although many studies and reviews have shown that *CDKN2A/B* deletion is associated with poor prognosis in diffuse glioma with *IDH1/2* mutation [3,11,12,18], relatively few have reported comprehensively on CNS WHO grade 4 gliomas. While managing CNS WHO grade 4 glioma patients in clinical practice, physicians encounter certain cases in which the prognosis of patients with *IDH*-mutant astrocytoma CNS WHO grade 4 is worse than that of patients with *IDH*-wildtype glioblastoma CNS WHO grade 4, even though *IDH* mutations are a good prognostic factor for gliomas. Therefore, we investigated if prognostic factors other than *IDH* mutation in patients with CNS WHO grade 4 gliomas are available. We hypothesized that *CDKN2A/B* deletion may play a similar role in CNS WHO grade 4 gliomas because it is a factor that deteriorates the prognosis of diffuse glioma, especially with *IDH* mutation.

Although *CDKN2A* deletion is well known to be a poor prognostic factor in low-grade diffuse gliomas with *IDH* mutations, the present study showed that it is also a poor prognostic factor in higher-grade gliomas that acts as a more important prognostic factor than *IDH* mutation, especially in CNS WHO grade 4 gliomas. More specifically, if patients with CNS WHO grade 4 glioma do not have *CDKN2A* deletion, they have longer PFS and OS than those with a *CDKN2A* deletion, regardless of the presence of *IDH* mutation. To the best of our knowledge, this is the first study to report that *CDKN2A* deletion acts as a more powerful prognostic factor than *IDH* mutation in WHO grade 4 CNS gliomas. Research on the role of *CDKN2A* deletion in patients with glioblastoma is rare [19,20]. One systematic review reported that glioblastoma patients had shorter PFS in the presence of *CDKN2A* homozygous deletion in two studies (median values, 16 vs. 30 months) and shorter OS in four studies (median values, 38 vs. 86 months) [11]. In multivariate analyses, *CDKN2A* homozygous deletion was a predictor of significantly shorter PFS and OS in low-grade glioma and glioblastoma across all included studies [11]. However, this study was published before the release of the new 2021 version of the WHO classification of CNS tumors and did not include classification of grade 4 glioma according to *IDH* mutation. Therefore, in the present study, the role of *CDKN2A* deletion according to *IDH*-mutation was not differentiated.

In this study, although patients with *CDKN2A* deletion had a slightly higher incidence of *TERT* promoter mutation, this trend did not have a mutually dependent effect on PFS and OS, and there was no other association between *CDKN2A* deletion and other genetic mutations when evaluating prognosis. Funakoshi et al. reported that *CDKN2A* homozygous deletion had no significant effect on OS in patients with a methylated MGMT gene promoter (*p* = 0.527), whereas among patients with an unmethylated MGMT gene promoter, there was a significant difference in OS between patients with and without *CDKN2A* homozygous deletion (*p* = 0.013) [19]. In addition, they showed that the difference was more evident in patients before treatment with bevacizumab (*p* = 0.035) but turned out to be non-significant in patients after treatment with bevacizumab (*p* = 0.101) due to OS improvement in patients with *CDKN2A* homozygous deletion [19]. However, these findings were not validated in The Cancer Genome Atlas cohort [19]. These results could not be confirmed in our study because the number of patients treated with bevacizumab was too small (27 patients, 19.9%) in the cohort of this study, which may have resulted in different research results from those of Funakoshi’s study. In fact, the present study showed that the presence or absence of *CDKN2A* deletion was not related to MGMT gene promoter methylation and was also independently associated with short PFS and OS regardless of MGMT gene promoter methylation (Appendix A).

In the present study, NGS analysis suggested that *CDKN2A* homozygous deletion was the sole mutation pattern observed in 75 samples with *CDKN2A* mutation. Although there are a variety of different types of mutations in *CDKN2A,* including hemizygous deletion, missense, nonsense, frame-shift mutations, etc., these mutations are rare compared to *CDKN2A* deletion; evaluation of large lower-grade *IDH*-mutant glioma cohorts demonstrates a mutation rate of 0.8–3.7%, while homozygous deletion occurs in approximately 7–20% of these otherwise histologically lower-grade gliomas [12,18,21,22,23,24,25]. Yokoda RT et al. illustrated that there is no significant difference in PFS or OS between grade-matched *IDH*-mutant astrocytomas with mutant or deleted *CDKN2A*. Furthermore, tumors with both these alterations appear to confer significantly worse PFS and OS compared to tumors with retained/wildtype *CDKN2A* in both univariate and multivariate models. Additionally, there is no apparent prognostic relevance for histologic grade in *IDH*-mutant astrocytoma with *CDKN2A* mutation/deletion unlike in those with wildtype *CDKN2A* [18]. Since homozygous *CDKN2A* deletion accounts for most *CDKN2A* mutations and has a prognosis equivalent to that of *IDH*-mutant astrocytoma with other *CDKN2A* mutations (and hemizygous *CDKN2A* loss), it can be used as a surrogate marker for determining prognosis [18].

Although we analyzed *CDKN2A* homozygous deletion using an NGS panel, it may be detected by a variety of methods, including fluorescence in situ hybridization (FISH), whole-exome sequencing, global DNA methylation profiling, and detecting potential loss of p16 immunoreactivity in tumor cell nuclei. Although FISH can detect many genomic alterations by simple morphological assessment and is widely used to detect copy number variations (CNVs), it is not able to differentiate between whole-arm and partial deletions, which are strongly associated with meaningful clinical outcomes [26]. In addition, in FISH, the assessment of deletions is somewhat subjective and sometimes challenging, owing to the presence of overlapping or partially sectioned nuclei. Therefore, diverse methods for detecting CNV have been developed for clinical purposes to overcome the limitations of FISH techniques, including comparative genomic hybridization [27], single-nucleotide polymorphism arrays [28], Infinium MethylationEPIC BeadChip DNA methylation arrays (EPIC array; Illumina) [29], and whole-genome sequencing (WGS), which is the most efficient platform for CNV detection [30]. These methods not only differentiate between partial or whole-arm chromosome gains/losses, but also detect other alterations such as *CDKN2A/B* deletion and epidermal growth factor receptor (EGFR) amplification [29,31]. Both array- and WGS-based techniques reduce the risk of false-positive results when testing for CNV detection. Deep- or low-coverage WGS data provided higher resolution and outperformed array-based detection [30,32]. However, these techniques are not as practical as clinical tests because the cost of WGS is still considerably high and special equipment is required.

As described above, various efforts are being made to overcome various limitations of existing diagnostic methods for detecting *CDKN2A* genetic status. Tian and colleagues [33] measured the copy number of the *CDKN2A* with the quantitative multiplex PCR assay P16-Light and validated results with WGS. They found 5.1 kb *CDKN2A* common deletion regions (CDR) in >90% of gastric cancers containing *CDKN2A* deletion. They suggested that *CDKN2A* CDR could be used as a potential target for developing the P16-Light assay to detect *CDKN2A* deletion and amplification for routine clinical practices. CDKN2A amplification and deletion have been reported to be considered potential biomarkers for predicting the regression of precancers in esophageal squamous cells and risk-stratifying patients with esophageal dysplasia [34,35]. However, *CDKN2A* amplification was not found in the NGS analysis result of CNS WHO grade 4 glioma. It is speculated that these brain tumors have a different pathophysiology from those of gastric and esophageal cancer, and they may not have been found because the number of samples of CNS WHO grade 4 glioma used in this study was relatively small. Comprehensive and extensive research using a larger number of samples is needed in the future.

As another method of conducting practical tests for detecting *CDKN2A* deletions, immunohistochemical staining for p16 can be considered a substitute for array-based or WGS-based techniques. As described in the introduction, the *CDKN2A* gene product is the p16 protein, and immunohistochemical detection of p16 protein expression can be used instead of molecular testing to identify *CDKN2A* gene deletions [36,37,38]. In the same way, Wakabayashi et al. have published a study showing that the hypermethylation of the *p16* gene rather than the p16 protein in astrocytoma is associated with the loss of function of the *CDKN2A* gene, and through this, the prognosis of glioma can be predicted [39]. They showed that similar methylations were detected in the serum of patients with aberrant methylation in the tumor tissues, but no methylated p16 sequences were detected in the peripheral serum of the patients with tumors without these methylation changes or in the healthy controls [39].

There is growing evidence suggesting the importance of *CDKN2A* deletion as a prognostic marker for adverse clinical outcomes in several CNS tumors, including supratentorial ependymoma with ZFTA fusion [40], high-grade meningioma [41], anaplastic *IDH*-mutant astrocytoma, and oligodendroglioma [22]. However, no comprehensive studies have shown the pathophysiological mechanism of *CDKN2A* deletion in the poor outcomes of CNS tumors. In the present study, clinical data analysis did not confirm a significant correlation between *CDKN2A* deletion and other gene mutations in CNS WHO grade 4 gliomas. The *CDKN2A* locus on chromosome band 9p21 encodes for two tumor suppressors, protein p14^ARF^ and p16^INK4A^, both inhibiting cell cycle progression [42,43]. However, little is known about the mechanisms by which these cell cycle regulators induce biological reactions and affect the prognosis of patients with CNS tumors. Some reports have suggested that postoperative irradiation increases the risk of *CDKN2A* homozygous deletion, which is frequently observed in recurrent grade 4 *IDH*-mutant astrocytomas and is associated with increased cellular proliferation [44,45]. Postoperative chemoradiotherapy combined with temozolomide was associated with increased *CDKN2A* deletion, possibly due to the higher selection pressure to inactivate the pathway in this treatment setting than after postoperative radiation alone [46]. Histopathologically, Appay et al. showed that among *IDH*-mutant gliomas, the presence of *CDKN2A* deletion was associated with a higher mitotic count and Ki67 labeling index than in those without *CDKN2A* deletion [12]. Another recent study focused on metabolic heterogeneity in glioblastoma. Minami et al. integrated lipidomic, transcriptomic, and genomic profiling data to identify altered lipid metabolism in glioblastoma with *CDKN2A* deletion. They suggested that *CDKN2A* deletion could remodel the distribution of polyunsaturated fatty acids into different lipid compartments, sensitizing glioblastomas with *CDKN2A* loss to lipid peroxidation and ferroptosis both in vitro and in vivo [47].

Interestingly, our study showed that *CDKN2A* homozygous deletion played a significant role as a negative prognostic factor in CNS WHO grade 4 glioma patients, regardless *of the IDH* mutation status. *CDKN2A* deletion has a stronger predictive power for prognosis than *IDH* mutation, which is a well-known traditional player in the prognosis of patients with glioma. However, there are major limitations to our study, such as lack of external data to validate the present results. This study was not a multi-center participation study, but involved a cohort treated in single institute. It was also not a randomized clinical trial (RCT) through prospective study design. Although our data are homogeneous and unique because they were obtained from the single tertiary-level university hospital covering a province of 1.2 million people followed up for a long time by the same treatment protocol and a uniform system, they are bound to have limitations compared to research results obtained through a randomized clinical trial. As the present study is not an RCT study, external validation is essential to prove the value of the study. External validation using public genetic information such as actual The Cancer Genome Atlas (TCGA) data was not conducted. Also, validation using high-level evidence such as meta-analysis is mandatory. To the best of our knowledge, there has been no published meta-analysis or systemic review focused on the prognostic role of *CDKN2A/B* deletion in entire CNS WHO grade 4 gliomas with or without *IDH* mutation. Therefore, the conclusions drawn from our study require further validation in prospective randomized clinical trials and meta-analysis including multiple studies on the prognostic role of *CDKN2A/B* deletion in CNS WHO grade 4 gliomas.

## 5. Conclusions

In the present study, we investigated the prognostic role of *CDKN2A* homozygous deletion in WHO grade 4 CNS gliomas, including *IDH*-mutant astrocytoma’s and *IDH*-wildtype glioblastomas. We found that *CDKN2A* homozygous deletion strongly affected the prognosis of WHO grade 4 glioma patients with *IDH* mutation as well as the *IDH* wildtype. In addition, this study suggests that *CDKN2A* homozygous deletion should be stronger than *IDH* mutation in the prognosis of CNS WHO grade 4 glioma patients. However, several practical difficulties exist in the clinical application of genetic and analytical methods for detecting *CDKN2A* mutations in glioma research. The lack of evidence showing the biological mechanism of *CDKN2A* deletion in affecting the prognosis of patients with CNS WHO grade 4 glioma makes a comprehensive study mandatory.

## Figures and Tables

**Table 1 biomedicines-12-02256-t001:** Clinical and genetic characteristics of whole CNS WHO grade 4 glioma cohorts (n = 136).

Variables		Number
Age (years)	<50	47 (34.6%)
	≥50	89 (65.4%)
Sex	Male	84 (61.8%)
	Female	52 (38.2%)
WHO performance status	0	55 (40.4%)
	1	65 (47.8%)
	2	16 (11.8%)
Extent of resection	Biopsy	17 (12.5%)
	Subtotal resection	52 (38.2%)
	Gross total resection	67 (49.3%)
RPA class	III	32 (23.5%)
	IV	79 (58.1%)
	V	25 (18.4%)
*MGMT* gene promoter	Methylated	86 (63.2%)
	Unmethylated	50 (36.8%)
EGFR amplification	Yes	95 (69.9%)
	No	41 (30.1%)
*TERT* promoter mutation	Yes	71 (52.2%)
	No	65 (47.8%)
*CDKN2A* deletion	Yes	75 (55.1%)
	No	61 (44.9%)
*IDH* mutation	Yes	29 (21.3%)
	No	107 (78.7%)
Postoperative adjuvant therapy		
RTx and/or nitrosourea chemotherapy		35 (25.7%)
CCRT with temozolomide		101 (74.3%)
Salvage treatment after progression *		
Second surgical resection		74 (54.4%)
Repeated irradiation		63 (46.3%)
Salvage chemotherapy		83 (61.0%)
Supportive treatment only		17 (12.5%)

Abbreviations: CCRT, concurrent chemoradiotherapy; CDKN2A, cyclin-dependent kinase inhibitor 2A; EGFR, epidermal growth factor receptor; IDH, isocitrate dehydrogenase; *MGMT*, O6-methyl guanine DNA methyltransferase; RPA, recursive partitioning analysis; RTx, radiotherapy; TERT, telomerase reverse transcriptase; WHO, World Health Organization. * Some patients were treated with more than one modality.

**Table 2 biomedicines-12-02256-t002:** Clinical characteristics of CNS WHO grade 4 glioma cohorts according to *IDH* mutation (n = 136).

Variables		IDH Mutant (n = 29)	IDH Wildtype (n = 107)	*p* Value
Age (years)	<50	14 (48.3%)	33 (30.8%)	0.153
	≥50	15 (51.7%)	74 (69.2%)	
Sex	Male	17 (58.6%)	67 (62.6%)	0.772
	Female	12 (31.4%)	40 (37.4%)	
WHO performance status	0	11 (37.9%)	44 (41.1%)	0.806
	1	13 (44.8%)	52 (48.6%)	
	2	5 (17.3%)	11 (10.3%)	
Extent of resection	Biopsy	3 (10.3%)	14 (13.1%)	0.734
	Subtotal resection	10 (34.5%)	42 (39.2%)	
	Gross total resection	16 (55.2%)	51 (47.7%)	
RPA class	III	9 (31.0%)	23 (21.5%)	0.183
	IV	17 (58.6%)	62 (57.9%)	
	V	3 (10.4%)	22 (20.6%)	
*MGMT* gene promoter	Methylated	20 (69.0%)	66 (61.7%)	0.271
	Unmethylated	9 (31.0%)	41 (38.3%)	
EGFR amplification	Yes	18 (62.1%)	71 (66.4%)	0.562
	No	11 (37.9%)	36 (33.6%)	
*TERT* promoter mutation	Yes	15 (51.7%)	56 (52.3%)	0.933
	No	14 (48.3%)	51 (47.7%)	
*CDKN2A* deletion	Yes	16 (55.2%)	59 (55.1%)	0.958
	No	13 (44.8%)	48 (44.9%)	
Postoperative adjuvant therapy				
RTx and/or nitrosourea chemotherapy		7 (24.1%)	28 (26.2%)	0.893
CCRT with temozolomide		22 (75.9%)	79 (73.8%)	
Salvage treatment after progression *				
Second surgical resection		15 (51.7%)	59 (55.1%)	0.725
Repeated irradiation		11 (37.9%)	52 (48.6%)	
Salvage chemotherapy		17 (58.6%)	66 (61.7%)	
Supportive treatment only		3 (10.3%)	14 (13.1%)	

Abbreviations: CCRT, concurrent chemoradiotherapy; *CDKN2A*, cyclin-dependent kinase inhibitor 2A; EGFR, epidermal growth factor receptor; *IDH*, isocitrate dehydrogenase; *MGMT*, O6-methyl guanine DNA methyltransferase; RPA, recursive partitioning analysis; RTx, radiotherapy; *TERT*, telomerase reverse transcriptase; WHO, World Health Organization. * Some patients were treated with more than one modality.

**Table 3 biomedicines-12-02256-t003:** Clinical characteristics of CNS WHO grade 4 glioma cohorts according to *CDKN2A* deletion status (n = 136).

Variables		*CDKN2A* Deletion (n = 75)	*CDKN2A* Intact (n = 61)	*p* Value
Age (years)	<50	21 (28.0%)	26 (42.6%)	0.054
	≥50	54 (72.0%)	35 (57.4%)	
Sex	Male	47 (62.7%)	37 (60.7%)	0.872
	Female	28 (37.3%)	24 (39.3%)	
WHO performance status	0	29 (38.7%)	26 (42.6%)	0.626
	1	36 (48.0%)	29 (47.5%)	
	2	10 (13.3%)	6 (9.9%)	
Extent of resection	Biopsy	10 (13.3%)	7 (11.5%)	0.791
	Subtotal resection	29 (38.7%)	23 (37.7%)	
	Gross total resection	36 (48.0%)	31 (50.8%)	
RPA class	III	16 (21.3%)	16 (26.2%)	0.617
	IV	43 (57.3%)	36 (59.0%)	
	V	16 (18.4%)	9 (14.8%)	
*MGMT* gene promoter	Methylated	47 (62.7%)	39 (61.9%)	0.935
	Unmethylated	28 (37.3%)	22 (36.1%)	
EGFR amplification	Yes	51 (68.0%)	44 (72.1%)	0.804
	No	24 (31.0%)	17 (27.9%)	
*TERT* promoter mutation	Yes	44 (58.7%)	27 (44.3%)	0.103
	No	31 (41.3%)	34 (55.7%)	
*IDH* mutation	Yes	16 (21.3%)	13 (21.3%)	0.958
	No	59 (78.7%)	48 (78.7%)	
Postoperative adjuvant therapy				0.454
RTx and/or nitrosourea chemotherapy		16 (21.3%)	19 (31.1%)	
CCRT with temozolomide		59 (78.7%)	42 (68.9%)	
Salvage treatment after progression *				0.836
Second surgical resection		38 (50.7%)	36 (59.0%)	
Repeated irradiation		32 (42.7%)	31 (50.8%)	
Salvage chemotherapy		46 (61.3%)	37 (60.7%)	
Supportive treatment only		10 (13.3%)	7 (11.5%)	

Abbreviations: CCRT, concurrent chemoradiotherapy; CDKN2A, cyclin-dependent kinase inhibitor 2A; EGFR, epidermal growth factor receptor; IDH, isocitrate dehydrogenase; *MGMT*, O6-methyl guanine DNA methyltransferase; RPA, recursive partitioning analysis; RTx, radiotherapy; TERT, telomerase reverse transcriptase; WHO, World Health Organization. * Some patients were treated with more than one modality.

**Table 4 biomedicines-12-02256-t004:** Univariate analysis of factors predicting progression-free survival (PFS) and overall survival (OS) in CNS WHO grade 4 glioma cohorts using Cox regression model according to the clinical factors.

Variables		Mean PFS (Month, ±SD)	Hazard Ratio (95% CI)	*p*-Value	Mean OS (Month, ±SD)	Hazard Ratio (95% CI)	*p*-Value
Age (years)	≥50	9.96 (±0.400)			18.91 (±1.452)		
	<50	11.29 (±0.546)	3.072 (0.967–5.177)	0.080	24.37 (±2.014)	6.326 (4.228–8.424)	0.012
Sex	Male	10.07 (±0.513)			20.15 (±1.992)		
	Female	10.91 (±0.532)	2.310 (0.906–3.714)	0.129	21.55 (±2.127)	1.528 (0.827–2.229)	0.467
WHO performance	2	8.65 (±0.388)			10.26 (±0.726)		
	1	10.11 (±0.617)	2.252 (0.911–3.593)	0.133	19.53 (±1.538)	10.254 (6.925–13.583)	0.004
	0	11.23 (±0.638)	3.679 (0.982–6.376)	0.055	24.41 (±2.513)	22.012 (15.32–28.70)	<0.001
Extent of resection	Bx	5.95 (±0.325)			13.58 (±1.185)		
	STR	9.40 (±0.418)	9.514 (7.008–12.021)	0.002	19.52 (±1.627)	4.595 (1.889–7.301)	0.034
	GTR	12.24 (±0.629)	41.251 (34.897–47.605)	<0.001	23.13 (±2.229)	7.980 (4.669–11.291)	0.007
RPA class	V	8.97 (±0.524)			10.54 (±0.825)		
	IV	10.25 (±0.596)	6.288 (5.543–7.033)	0.012	20.05 (±1.797)	11.620 (7.074–16.166)	<0.001
	III	11.77 (±0.662)	8.648 (6.893–10.403)	0.003	29.36 (±2.554)	23.161 (17.43–28.90)	<0.001
*MGMT* gene promoter							
Unmethylated		9.31 (±0.503)			17.29 (±2.327)		
Methylated		11.02 (±0.672)	7.330 (6.134–8.526)	0.017	22.28 (±2.506)	6.306 (4.653–7.959)	0.008
EGFR amplification	Yes	9.49 (±0.603)			20.17 (±1.962)		
	No	12.39 (±0.758)	14.536 (12.085–16.987)	<0.001	22.03 (±1.994)	1.989 (0.857–3.121)	0.320
*TERT* mutation	Yes	9.33 (±0.567)			19.66 (±1.878)		
	No	11.53 (±0.706)	17.490 (14.922–20.058)	<0.001	21.87 (±1.999)	2.586 (0.929–4.243)	0.208
*CDKN2A* deletion	Yes	8.75 (±0.522)			16.65 (±1.689)		
	No	12.40 (±0.735)	33.218 (30.085–36.351)	<0.001	25.05 (±2.811)	11.129 (7.048–15.211)	0.002
*IDH* mutation	No	10.16 (±0.632)			19.12 (±2.227)		
	Yes	11.31 (±0.681)	3.349 (0.957–5.741)	0.067	26.02 (±2.513)	5.794 (3.185–8.403)	0.023
Postop. adjuvant therapy							
RTx and/or nitrosourea chemotherapy		10.38 (±0.533)			18.44 (±1.527)		
CCRT with TMZ		10.40 (±0.584)	1.018 (0.486–1.551)	0.984	21.65 (±1.869)	3.456 (0.970–5.942)	0.063

Abbreviations: Bx, biopsy; CCRT, concurrent chemoradiotherapy; *CDKN2A*, cyclin-dependent kinase inhibitor 2A; CI, confidence interval; EGFR, epidermal growth factor receptor; GTR, gross total resection; *IDH*, isocitrate dehydrogenase; *MGMT*, O6-methyl guanine DNA methyltransferase; PFS, progression-free survival; RPA, recursive partitioning analysis; RTx, radiotherapy; STR, subtotal resection; *TERT,* telomerase reverse transcriptase; SD, standard deviation; TMZ, temozolomide; WHO, World Health Organization.

**Table 5 biomedicines-12-02256-t005:** Univariate analysis of factors predicting progression-free survival (PFS) and overall survival (OS) in CNS WHO grade 4 glioma cohorts using Cox regression model focused on risk group.

Groups	Mean PFS (month, ±SD)	Hazard Ratio (95% CI)	*p*-Value	Mean OS (month, ±SD)	Hazard Ratio (95% CI)	*p*-Value
High-risk group (n = 75)	8.75 (±0.694)			16.65 (±1.554)		
Intermediate-risk group (n = 48)	11.02 (±0.956)	6.469 (5.048–7.889)	<0.001	21.85 (±2.082)	4.792 (2.015–7.569)	0.029
Low-risk group (n = 13)	15.25 (±1.492)	16.979 (13.650–20.308)	<0.001	33.38 (±2.946)	12.455 (9.627–15.283)	<0.001

Abbreviations: CI, confidence interval; OS, overall survival; PFS, progression-free survival; SD, standard deviation. High-risk group = *CDKN2A* deletion regardless of *IDH1/2* mutation/intermediate-risk group = *CDKN2A* intact + *IDH1/2* wildtype/low-risk group = *CDKN2A* intact + *IDH1/2* mutant.

**Table 6 biomedicines-12-02256-t006:** Multivariate analysis of factors predicting progression-free survival (PFS) and overall survival (OS) in CNS WHO grade 4 glioma cohorts using Cox regression model.

Variables	Progression-Free Survival		Overall Survival	
	Hazard Ratio (95% CI)	*p*-Value	Hazard Ratio (95% CI)	*p*-Value
Age (<50 years vs. ≥50 years)	2.652 (0.924–4.381)	0.164	4.645 (2.865–6.425)	0.037
WHO performance status (0 vs. 1)	1.728 (0.772–2.684)	0.465	3.817 (2.436–5.198)	0.046
(0 vs. 2)	2.890 (0.874–4.906)	0.359	5.002 (3.756–6.248)	0.039
(1 vs. 2)	1.637 (0.689–2.585)	0.518	3.663 (1.492–5.834)	0.049
Extent of surgery (GTR vs. biopsy)	11.651 (8.755–14.547)	<0.001	8.075 (5.837–10.313)	0.006
(GTR vs. STR)	9.323 (7.285–11.361)	0.003	5.528 (3.840–7.216)	0.030
(STR vs. Biopsy)	8.609 (6.238–10.979)	0.009	3.233 (0.982–5.484)	0.053
RPA class (III vs. IV)	3.408 (0.952–5.864)	0.078	3.992 (2.008–5.976)	0.045
(III vs. V)	5.382 (3.894–6.869)	0.028	6.773 (4.259–9.287)	0.022
(IV vs. V)	4.611 (1.996–7.226)	0.038	5.019 (3.890–6.148)	0.036
*MGMT* gene promoter (methylated vs. unmethylated)	6.989 (5.198–8.779)	0.018	5.078 (3.694–6.462)	0.030
EGFR amplification (No vs. Yes)	9.658 (8.113–11.203)	0.001	2.748 (0.825–4.671)	0.152
*TERT* promoter mutation (No vs. Yes)	13.077 (10.840–15.314)	<0.001	3.179 (0.887–5.471)	0.138
*CDKN2A* deletion (No vs. Yes)	21.361 (18.651–24.071)	<0.001	13.454 (10.268–16.639)	<0.001
*IDH* mutation (Yes vs. No)	3.085 (0.888–5.282)	0.116	6.352 (5.079–7.625)	0.028
Postoperative adjuvant therapy (RTx and/or nitrosourea vs. CCRT with TMZ)	1.373 (0.462–2.284)	0.995	2.237 (0.842–3.632)	0.195

Abbreviations: CCRT, concurrent chemoradiotherapy; *CDKN2A*, cyclin-dependent kinase inhibitor 2A; CI, confidence interval; EGFR, epidermal growth factor receptor; *IDH*, isocitrate dehydrogenase; GTR, gross total resection; *MGMT*, O6-methyl DNA guanine methyltransferase; RPA, recursive partitioning analysis; RTx, radiotherapy; STR, subtotal resection; TMZ, temozolomide; *TERT,* telomerase reverse transcriptase; WHO, World Health Organization.

## Data Availability

The data presented in this study are available upon request from the corresponding author. These data are not publicly available because of privacy restrictions, since they contain information that could compromise the privacy of the study participants.

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
