# Peer review of "CDKN2A Homozygous Deletion Is a Stronger Predictor of Outcome than IDH1/2-Mutation in CNS WHO Grade 4 Gliomas"

_biomedicines, 2024, doi:10.3390/biomedicines12102256_

Round 1

Reviewer 1 Report

Comments and Suggestions for Authors

The submitted manuscript is interesting and makes the necessary statistical analyses that support the theory that homozygous deletion of CDKN2A is a more effective poor prognostic factor than the IDH1/2 mutational status.

Minor recommendations:

Materials and Methods

Briefly explains the differences in RPA classes and in which types of patients this classification was applied

Results

In tables 4 – 8, and their respective figures in the figure caption, the number of patients is missing. This information is relevant to know the frequency of each of the groups A, B, C and D, as well as the categorization of the High, Intermediate and Low risk groups.

Discussion

The explanation of FISH in the detection of CDKN2A is redundant.

Author Response

The submitted manuscript is interesting and makes the necessary statistical analyses that support the theory that homozygous deletion of CDKN2A is a more effective poor prognostic factor than the IDH1/2 mutational status.

Authors’ response) First of all, we would like to thank you for your careful and precious review opinions. We did our best to revise the reviewer's point, and the revised part was marked in blue in the manuscript. Even if it is troublesome and troublesome, please review the revised manuscript again, and if there are still some incompleteness or additional revisions despite our efforts, please give us valuable opinions.

1. Materials and Methods. Briefly explains the differences in RPA classes and in which types of patients this classification was applied

Author’s response) As the reviewer pointed out, we inserted a brief description of the RPA class as below sentence.

“For example, patients with RPA class III are defined as those under the age of 50 with a KPS score of ≥90, and patients with RPA class IV are those under the age of 50 with a KPS score of <90, or those over the age of 50 with a KPS score of ≥70 and underwent gross total resection (GTR), and patients with RPA class V are defined as those over the age 50 with a KPS score of ≥70 but underwent subtotal resection (STR) or biopsy only, or those over the age of 50 and a KPS score of <70 [14].”

2. Results. In tables 4 – 8, and their respective figures in the figure caption, the number of patients is missing. This information is relevant to know the frequency of each of the groups A, B, C and D, as well as the categorization of the High, Intermediate and Low risk groups.

Author’s response) The number of patients for each variable is already shown in table 1. Of course, if the number of patients for all variables is marked on each table, it has a good advantage to refer simply, but there is a concern of repetition because it is repeatedly shown. The number of patients in the risk group which is a variable not shown in Table 1, is shown in the table 6 and 7 (table 5 in the revised manuscript). For the reasons explained above, please understand that the part reviewer pointed out were left unmodified.

3. Discussion. The explanation of FISH in the detection of CDKN2A is redundant.

Author’s response) Trying to explain why NGS technology was used rather than the most widely used FISH method in the clinical field seems to have led to excessive explanation of the disadvantages and limitations of the FISH method. As the reviewer pointed out, the sentences of literature reviews corresponding to reference #33, #34, and #36 have been deleted.

Reviewer 2 Report

Comments and Suggestions for Authors

In this manuscript the authors investigate the prognostic role of CDKN2A homozygous deletion in WHO- 80 grade 4 CNS gliomas.  

Overall, it is a clear, well-written manuscript that deserves to be published. I do not have any comments/questions about the study. 

Author Response

In this manuscript the authors investigate the prognostic role of CDKN2A homozygous deletion in WHO grade 4 CNS gliomas.  Overall, it is a clear, well-written manuscript that deserves to be published. I do not have any comments/questions about the study. 

Author’s response) We would like to say thank the reviewer’s compliment, and thank you for the grace comments.

Reviewer 3 Report

Comments and Suggestions for Authors

The CDKN2A/p16 gene is frequently inactivated by copy number deletion and DNA methylation in some cancer including glioma. Dr. Lee et al compared prognosis of WHO grade 4 glioma from 136 patients with a set of typical biomarkers including IDH mutation, MGMT methylation, and CDKN2A homozygous deletion. They found that the progression-free survival and overall survival of these patients were more strongly associated with CDKN2A deletion than IDH mutation and suggest a classification strategy the grade 4 glioma. Their findings are novel and useful for clinical management of the disease.

Main concerns:

1. There are two many unnecessary tables and figures. For example, Table 4  and Figure 1 present the same results; Table 5  and Figure 2 present the same results;  data in Table 4 and 5 can be merged and should not be displayed in Figure 3 and 4 one more time. Similar modifications should be made for Table 6 and 7 and Figure 5.

2. Overall survival is a more reliable endpoint than progression-free survival. It is not necessary to present so many statistical analysis results on progression-free survival in the Results section.

3. Multivariate analysis errors. Risk group classification is made according to the status of IDH mutation and CDKN2A deletion. To avoid double-adjusting, the risk factor should not been included in the multivariate analysis because both IDH mutation and CDKN2A deletion have been included in the analysis.

4. They classified these glioma samples as IDH mutation-positive and -negative according to target probe-captured NGS sequencing. There are many kinds of point mutation or microdeletion/insertion. The reviewer did not see any definition on "IDH mutation". At least it should be subclassified as nonsynonymous mutations and synonymous mutation (or wild type).

5. They detected the methylation status of the MGMT gene by methylation-specific PCR. The frequency of CDKN2A/p16 methylation in astrocytoma is approximate 60% (Wakabayashi et al. Neurosurgery 2009). It will greatly improve the quality of the manuscript if they could detect the prevalence of p16 methylation in these grade 4 glioma samples and analyze the efficiency of CDKN2A deletion combined with p16 methylation on the evaluation of glioma prognosis.

6. Recently, a Chinese research term reported a 5.1kb base-resolution common deletion region within the CDKN2A gene in cancer samples with CDKN2 deletion and setup a sensitive quantitative PCR assay called P16-Light (Tian et al. Frontiers Oncol 2022). Their studies further revealed that  both somatic copy number deletion and amplification of CDKN2A were prevalent in precancerous lesions and non-cancerous tissue surrounding cancer mass and that CDKN2A amplification is significantly associated with regression of esophageal squamous cell dysplasia (Fan et al. Chinese Med J 2024) and good prognosis of gastric carcinoma (Deng et al. Gastric Cancer 2024). It is  interesting whether CDKN2A amplification occurs in non-glioma brain tissue samples from patients received the gloss total resection treatment. At least they need to discuss these progressions.

7. There are many publicly available sequence dataset on the TCGA databases. The reviewer suggests the authors perform a meta analysis to validate their findings.

Comments on the Quality of English Language

Generally, people say the prediction of therapy outcomes, the evaluation of prognosis. Mean/average values are often accompanied with the standard deviation for the normally distributed data, and median values are often accompanied with the data range (full or 25-75% of range).

Some sentences are difficult to follow.

Author Response

The CDKN2A/p16 gene is frequently inactivated by copy number deletion and DNA methylation in some cancer including glioma. Dr. Lee et al compared prognosis of WHO grade 4 glioma from 136 patients with a set of typical biomarkers including IDH mutation, MGMT methylation, and CDKN2A homozygous deletion. They found that the progression-free survival and overall survival of these patients were more strongly associated with CDKN2A deletion than IDH mutation and suggest a classification strategy the grade 4 glioma. Their findings are novel and useful for clinical management of the disease.

Authors’ response) First of all, we would like to thank you for your careful and precious review opinions. We did our best to revise the reviewer's point, and the revised part was marked in red in the manuscript. Even if it is troublesome and troublesome, please review the revised manuscript again, and if there are still some incompleteness or additional revisions despite our efforts, please give us valuable opinions.

1. There are too many unnecessary tables and figures. For example, Table 4 and Figure 1 present the same results; Table 5 and Figure 2 present the same results; data in Table 4 and 5 can be merged and should not be displayed in Figure 3 and 4 one more time. Similar modifications should be made for Table 6 and 7 and Figure 5.

Author’s Response) As the reviewer pointed out, the tables and figures that show the same results repeatedly are separated, leaving only the table, and all the figures are separated into supplementary files and processed so that they are not visible in the text.

In addition, PFS and OS were described in different tables, and they were merged and displayed as one table. For example, Tables 4 and 5, Tables 6 and 7, and Tables 8 and 9 were merged into one respectively.

Author’s Response) We put together tables showing PFS and OS as recommended by the reviewer. For example, the original tables 4 and 5 are Table 4 in revised manuscript, the original tables 6 and 7 are Table 5 in revised manuscript, and the original tables 8 and 9 are Table 6 in revised manuscript.

2. Overall survival is a more reliable endpoint than progression-free survival. It is not necessary to present so many statistical analysis results on progression-free survival in the Results section.

Author’s Response)  In the result section, only descriptions of factors that are statistically meaningful to PFS are left, and additional descriptions of factors that are not related to PFS are deleted (subtitle 3.2 and 3.5)

3. Multivariate analysis errors. Risk group classification is made according to the status of IDH mutation and CDKN2A deletion. To avoid double-adjusting, the risk factor should not be included in the multivariate analysis because both IDH mutation and CDKN2A deletion have been included in the analysis.

Author’s Response)  Thank you for pointing out an important statistical error that we didn't think of. As you pointed out, the risk group was excluded from the multivariate analysis to avoid the error of overlapping analysis of IDH mutation and CDKN2A deletion. As a result, Hazard ratio and confidence interval was changed minimally, and the changed values were re-stated in the table 6 of revised version. However, it is fortunate that there was no variable with p-value change.

4. They classified these glioma samples as IDH mutation-positive and -negative according to target probe-captured NGS sequencing. There are many kinds of point mutation or microdeletion/insertion. The reviewer did not see any definition on "IDH mutation". At least it should be subclassified as nonsynonymous mutations and synonymous mutation (or wild type).

Author’s Response)  Thank you for your detailed information. Despite the basic content, describing the definition of IDH-mutation was missing. In this study, as mentioned in the materials and methods, IDH-mutation was detected through NGS analysis. Out of 29 patients with IDH-mutation, only 1 patient had IDH2 mutation, and all others had IDH1 mutation. A sentence describing the definition of IDH mutation was inserted into the corresponding section of materials and methods as follows. The number of patients with IDH1 and 2 mutation was already described in the result [Line 172-175 in the revised manuscript].

“The IDH-mutation was defined in this analysis as single amino acid missense mutation in IDH1 at arginine 132 (R132) or the analogous residue in IDH2 (R140 or R172) by the single-nucleotide polymorphism (SNP) and insertion/deletion polymorphism (INDEL) analysis of NGS.”

5. They detected the methylation status of the MGMT gene by methylation-specific PCR. The frequency of CDKN2A/p16 methylation in astrocytoma is approximate 60% (Wakabayashi et al. Neurosurgery 2009). It will greatly improve the quality of the manuscript if they could detect the prevalence of p16 methylation in these grade 4 glioma samples and analyze the efficiency of CDKN2A deletion combined with p16 methylation on the evaluation of glioma prognosis.

Author’s Response) Thank you for your great comment. I wish I were able to show the results by performing methylation analysis on the p16 gene. But we need to supplement the overall research process such as re-experiment and revision of the research design, so we are sorry that we cannot do this experiment again in various limitations. Therefore, we inserted the description of your comments and references into the discussion as follows [Line 485-492 in the revised manuscript]:

“In the same way, Wakabayashi, et al. have published a study that the hypermethylation of the p16 gene rather than p16 protein in astrocytoma is associated with the loss of function of the CDKN2A gene, and through this, the prognosis of glioma can be predicted [36]. They showed that similar methylations were detected in the serum of the patients with aberrant methylation in the tumor tissues, but no methylated p16 sequences were detected in the peripheral serum of the patients having tumors without these methylation changes or in the healthy controls [36].”

6. Recently, a Chinese research term reported a 5.1kb base-resolution common deletion region within the CDKN2A gene in cancer samples with CDKN2 deletion and setup a sensitive quantitative PCR assay called P16-Light (Tian et al. Frontiers Oncol 2022). Their studies further revealed that both somatic copy number deletion and amplification of CDKN2A were prevalent in precancerous lesions and non-cancerous tissue surrounding cancer mass and that CDKN2A amplification is significantly associated with regression of esophageal squamous cell dysplasia (Fan et al. Chinese Med J 2024) and good prognosis of gastric carcinoma (Deng et al. Gastric Cancer 2024). It is interesting whether CDKN2A amplification occurs in non-glioma brain tissue samples from patients received the gloss total resection treatment. At least they need to discuss these progressions.

Author’s Response) Thank you for providing great information. As you requested, we reviewed the NGS result again to see if there is CDKN2A amplification for CNS WHO grade 4 glioma .However, there was no case with CDKN2A amplification in the sample of this study. Maybe this is because of the relatively small number of samples, and it is possible that they actually have a different pathophysiology from stomach cancer or esophageal cancer. We made a paragraph with these contents and described it in the discussion section [Line 465-480 in the revised manuscript].

7. There are many publicly available sequence dataset on the TCGA databases. The reviewer suggests the authors perform a meta-analysis to validate their findings.

Author’s Response) Thank you for your excellent suggestion. I completely agree with the excellent opinions of the reviewer that can improve the quality of the paper if I can revise the manuscript as reviewer’s comment. As you suggested, it would be nice to use the TCGA data set for external validation, but it is difficult to implement external validation due to various practical limitations. In addition, if there is a meta-analysis that analyzed the similar results, it can be compared with our results, but according to what we have found, there are a few meta-analysis on the role of CDKN2A/B deletion for IDH-mutant glioma, but no meta-analysis for the entire CNS WHO grade 4 glioma with or without IDH-mutation can be found. Therefore, this explanation was added at the end of the Discussion to describe the limitations of this study [Line 533-549 in the revised manuscript].

8. Generally, people say the prediction of therapy outcomes, the evaluation of prognosis. Mean/average values are often accompanied with the standard deviation for the normally distributed data, and median values are often accompanied with the data range (full or 25-75% of range).

Author’s Response) Thank you very much for your valuable comments. Although the authors wrote and presented more than 100 papers, no reviewers have pointed out this specific matter. Even the statistician who advises our statistical analysis has never raised a question with these errors, but thank you very much for making such an important comment. As you mentioned, all the contents indicating the mean values were revised to express standard deviation instead of 95% confidence interval, and Tables 5 and 6 and the parts corresponding to the text of the manuscript were also revised.

Round 2

Reviewer 3 Report

Comments and Suggestions for Authors

The authors addressed most of my concerns. Please describe how MGMT methylation was analyzed in the Method section.

Author Response

The authors addressed most of my concerns. Please describe how MGMT methylation was analyzed in the Method section.

Authors’ response) First of all, we would like to thank you for your detail review opinions. The NGS analysis panel, ONCOaccuPanel, which was used by our institution did not include MGTM gene alteration, so we did not investigate the methylation status of the MGMT gene promoter by NGS analysis. Our institution quantitatively analyzes the methylation status of the MGMT gene promoter using pyrosequencing, and as the research results were published in 2011 [15], MGMT gene promoter methylation has been analyzed using the pyrosequencing technique. The sentence for the methylation of the MGMT gene promoter using pyrosequencing was described in the subtitle of 2.2 clinical data (Line 129-134) as follows;

“MGMT gene promoter methylation has been tested using the pyrosequencing method as quantitative tool in our institute [15] instead of methylation specific polymerase chain re-action (PCR). The main analyses converted individual C/T ratio data into mean values of the five CpGs analyzed in a gene segment. The percentage of the mean value of five CpGs were considered methylated if the percentage was >9%, which was widely used reference in literatures [15].”